# Precision Phenotyping of Nectar-Related Traits Using X-ray Micro Computed Tomography

**DOI:** 10.3390/cells11213452

**Published:** 2022-10-31

**Authors:** Laurent Begot, Filip Slavkovic, Myriam Oger, Clement Pichot, Halima Morin, Adnane Boualem, Anne-Laure Favier, Abdelhafid Bendahmane

**Affiliations:** 1Imagery Unit, Institut de Recherche Biomédicale des Armées (IRBA), 91220 Brétigny-sur-Orge, France; 2Institute of Plant Sciences Paris-Saclay (IPS2), INRAE, CNRS, University of Paris-Saclay, Bâtiment 630, 91192 Gif sur Yvette, France

**Keywords:** *Cucumis melo*, flower, honey bee, image analysis, nectar, nectary, pollinators, X-ray microtomography (micro-CT)

## Abstract

Flower morphologies shape the accessibility to nectar and pollen, two major traits that determine plant–pollinator interactions and reproductive success. Melon is an economically important crop whose reproduction is completely pollinator-dependent and, as such, is a valuable model for studying crop-ecological functions. High-resolution imaging techniques, such as micro-computed tomography (micro-CT), have recently become popular for phenotyping in plant science. Here, we implemented micro-CT to study floral morphology and honey bees in the context of nectar-related traits without a sample preparation to improve the phenotyping precision and quality. We generated high-quality 3D models of melon male and female flowers and compared the geometric measures. Micro-CT allowed for a relatively easy and rapid generation of 3D volumetric data on nectar, nectary, flower, and honey bee body sizes. A comparative analysis of male and female flowers showed a strong positive correlation between the nectar gland volume and the volume of the secreted nectar. We modeled the nectar level inside the flower and reconstructed a 3D model of the accessibility by honey bees. By combining data on flower morphology, the honey bee size and nectar volume, this protocol can be used to assess the flower accessibility to pollinators in a high resolution, and can readily carry out genotypes comparative analysis to identify nectar-pollination-related traits.

## 1. Introduction

Flower morphology, together with color and scent, is one of the key traits that contribute to pollinator attraction, pollen dispersal, and plant reproductive success [1,2,3]. Flower morphology determines not only the physical access to nectar, but also the pollen deposition efficiency on the bee body and the acquisition of pollen by the stigma [4,5]. One of the best illustrating examples of an adaptation is the flower tube length in a hawkmoth-pollinated *Gladiolus* (Iridaceae) that positively correlated with the fruit and seed set production due to a better ‘fit’ between the flower and pollinator morphology [6].

Within the flower, specialized organs, nectaries, synthesize nectar in a process that is under complex developmental control [7]. At anthesis, the timing of nectar secretion coincides with the pollen release and the nectar flow is often increased by pollinator visitation, but ceases after pollination [8]. Therefore, nectary gland maturation, the duration of anthesis, and accessibility to the pollinators are all coordinated to ensure a successful and timely visit. It is well established for large number of species that there is a considerable intraspecific variability in nectar production. From the breeder’s perspective, the selection of traits that can improve the nectar-related qualities with consequences on the pollination efficiency could be key breeding targets. A correlation between the flower features, nectar volumes, and efficiency of pollination was reported in some species [9,10]. In *Medicago sativa*, a strong positive correlation between the diameter of the nectar reservoir and the nectar volume per floret was described [11]. *Brassica* species of varied ploidy levels also showed a positive correlation between the nectary size and nectar volume [12]. This could imply that a genetic selection should be directed towards the varieties with large organ sizes, large nectar reservoirs, or other flower features that correlate with the nectar quality or quantity-related traits [5,13]. Quantitative data on such traits in relation to the pollination efficiency, on the other hand, is scarce. The kinetics of nectar secretion are also not consistent throughout the day, making quantification more difficult. In *Cucurbita pepo*, nectar secretion is dependent on the daytime and the sex of the flower. Some plants also have the ability to reabsorb most or all of the unconsumed nectar [14,15]. Nevertheless, when discussing the floral traits which are suitable for breeding, it was suggested that any character that enhances plant–pollinator interactions and concurrently bears a high genetic variance is a potential target [16,17]. 

Traditionally, the phenotypic analyses of floral shape have been performed by classic morphometrics [13,18,19,20,21,22] which utilize multivariate statistics to measure the distances between anatomical landmarks (keypoints) [23]. Today, a morpho-anatomical analyses of any biological specimen can be performed using a combination of different methods, including the ones based on contour detection with Fourier analysis [24]: elliptic Fourier analysis [25], object detection [26], or the landmark-based method of geometric morphometrics [27,28]. The latter can be fully automated by using softwares such as *FlowerMorphology,* optimized for five-petal actinomorphic flowers [29]. In addition to 2D imaging, advances in image analysis allowed the 3D acquisition of the volumetric data suitable for the visualization, analysis, and quantification of the scanning results at a cellular and submicron resolution [30,31]. 

High-resolution imaging techniques such as X-ray computed tomography (micro-CT) have become increasingly popular for phenotypic trait measurements in plant science [32,33,34,35,36,37]. As a minimally invasive technique, micro-CT allows for the 3D reconstruction of scanned objects [32]. In plants, micro-CT had been used to dissect the plant anatomy of woody samples [38] and to measure the spatial distribution of trichomes on Arabidopsis leaves [39], as well as to measure the intracellular spaces in plant tissues [40,41]. Moreover, it has been used to quantify the different aspects of plant development, including ovules and pollen [42], the subcellular features of seeds [40], and fruit tissue [41]. Regarding pollinators, micro-CT was used to scan wasp and honey bee morphology, including the stingers [43] and pupae [44]. Taking these reports together, this technique could have a promising potential to improve the precision and phenotyping quality in the analysis of nectar-related traits.

Melon (*Cucumis melo*) is a eudicot diploid (2n = 24) species from the Cucurbitaceae family of economic and scientific value [45]. Like other Cucurbitaceae members, melon displays a range of sexual phenotypes, including various combinations of male, female, or bisexual flowers [46]. In contrast to most hermaphrodite plants, melon develops unisexual flowers and is strictly entomophilous, which makes fertilization completely pollinator-dependent [47]. For this reason, melon is also a valuable model for studying flower–pollinator interactions. 

In the present study, we used micro-CT to reconstruct the shape of the respective male and female melon flowers in the context of nectar biology. In most of the previous studies, the scanned tissues were previously fixed, which required an additional preparation time and may affect the shapes of the organs, especially the petals. The objective of this study was to optimize a protocol for the imaging of the non-fixed flowers to minimize the impact on the integrity of the shapes and volumes of the floral organs. We discuss the suitability of our protocol in the investigation of flower traits in relationship to a pollinator’s visitation. 

## 2. Materials and Methods

The mind map of the methodology used in this study is illustrated in Figure 1.

### 2.1. Plant Material and Sampling

Melon (*Cucumis melo*) plants of the Charentais mono and Charentais andromono variety were grown in a greenhouse (27/21°C, 40–50% RH, natural light supplemented with sodium high pressure lamps (PLANTASTAR 400 W), under a 16 h day/8 h night) of the Institute of Plant Sciences Paris Saclay in Gif-sur-Yvette, France. Male and female flowers were harvested at the anthesis stage. Nectar was removed from the flowers using glass capillaries (See Nectar volume measurement) and was recorded. After harvesting, the samples were covered with wet paper, closed in Petri dishes, and transported to the lab on ice. Individual flowers were placed by inserting the peduncle into a microtube through an open aperture in the cap. The microtube was filled with water to avoid dehydration during scanning.

### 2.2. Micro-CT Scanning Conditions

The microtomography used in this study is Bruker SkyScan 1276 [48,49]. The Bruker 1276 machine has a range of tension included between 20 to 100 kV, 20 W, and a spatial resolution of 2.8 µm for the smallest pixel size. To scan the melon flowers, we used the following parameters: source voltage = 30 kV; source current = 150 µA; rotation step = 0.2°; rotation scan = 180°; scaled image pixel size = 5 µm; and duration scan between 21 and 33 min [31,50]. For honey bees, the scanning parameters were as follows: source voltage = 54 kV; source current = 127 µA; rotation step = 0.4°; rotation scan = 360°; scaled image pixel size = 5 µm; and scan duration 23 min [40].

### 2.3. Reconstruction of 3D Images

All the resulting projection images were reconstructed using the NRecon software (v.1.7.3.0, Bruker-micro-CT, Kontich, Belgium) with post-alignment, beam hardening corrections (41%), and a ring artifact reduction (8). Three-dimensional images were cropped, rotated, and registered using the Data Viewer program (v.1.5.6.2 64 bit, Bruker-micro-CT, Kontich, Belgium) in order to optimize and facilitate the measurements on the different data sets.

### 2.4. Honey Bee Preparation

European honey bee *(Apis mellifera)* specimens were collected alive from a beehive in Brétigny-sur-Orge, France, and were prepared as described in [51,52]. Briefly, insects were conserved for 5 days in 70% ethanol, transferred in 1% iodine solution (Lugol) in 100% ethanol for 24 h before submersion in hexamethyldisilazane for 9–24 h, and were air-dried overnight. The honey bee was positioned in the scanning tube and covered by two half-circles of polystyrene in order to reduce the impact of the air pulse generated by the fan inside the microtomography instrument during the scan.

### 2.5. Nectary Cross-Sectional Area Measurements

Entire flowers were collected at the anthesis stage, longitudinally dissected and photographed using a Carl Zeiss Stereomicroscope 305. The nectary size (µm^2^) was measured using the ZEN 2.3 Lite Software by selecting the region of the nectary tissue (µm^2^) which was converted into mm^2^. In the case of the 3D micro-CT images, two perpendicular virtual longitudinal sections were made at the center of the flower. The nectary contours were automatically selected using FIJI software to calculate the mean (in mm²) of the organ area. 

### 2.6. Computation of Nectary Surface and Volume

Image processing tools, programmed in Python, were used on each micro-CT dataset in order to virtually extract the 3D structure of the nectary. The first step is the creation of a volume of interest starting from the first slide to the last slide where the 3D delimitation of the nectary area is observed. A threshold is then used in order to separate the voxels from the cellular tissues of the flower and those from the air. A Z-projection of the height of each element is generated, with the grey level corroborating with the height of the elements. Watershed segmentation is then used to obtain the contours of the compartments. The external contours of the nectary are selected using the threshold. The lower contours are computed by interpolation from the external contours, since they could not be obtained by the threshold. A nectary 3D image is generated and used to calculate the volume of the nectary. The pixels corresponding to the surface of gland in contact with air are then kept in order to measure the nectary surface. The image processing pipeline is deposited in the GitHub at https://github.com/El-Castor/X-ray-Micro-computed-Tomography (accessed on 20 September 2022).

### 2.7. Nectar Volume Measurements

We measured the nectar volume using the classical glass capillary method [53,54]. The nectar extraction from the flower was performed at the anthesis using 10 microliter capillary pipettes (Hirschmann REF 9600210) in the afternoon between 14 h and 15 h. The glass capillary was inserted between the petals and the stamens/carpels of intact flowers and nectar is taken up by the capillary force. The volume is measured by reading the length of the liquid column. Each sampled flower (n) represented a separate individual. The number of samples measured was 11 for male and 9 for female flowers.

### 2.8. Statistical Analysis

All statistical analyses were performed with the RStudio software, except for the linear regression and correlation analyses, which were done in Excel. The RStudio packages used are: tidyverse, ggpubr, outliers, and rstatix. For the comparison of male and female samples, the means of four different parameters were analyzed: the flower width, nectary cross-sectional area, nectary surface area, and volume values. Each population was tested with a Shapiro test to check if it follows a normal statistical distribution. A second step consisted of carrying out a Levene’s test, which determined if the population variances were equal or unequal. All Shapiro tests demonstrated that the samples are parametric. The Levene’s test result revealed equal variances between the samples for the parametric values for two flower parameters: the width and cross-sectional area, for which a Student’s *t*-test was applied. For the two other parameters (the nectary volume and surface area), the Levene’s test result showed unequal variances between the groups. A Welsh’s *t*-test was then done.

For the comparison of the nectar’s volume which was pipetted, the groups were tested two by two (M1 versus M2: for the intact and dissected male, and F1 versus F2: the intact and dissected female flowers) with Shapiro and Levene tests being followed by a Student’s *t*-test. 

The statistical graphic results of the averages comparison was classically represented with an asterisk, which is characteristic of the level of significance of the *p*-value: * = *p* < 0.05, ** = *p* < 0.01, *** = *p* < 0.001.

## 3. Results

### 3.1. Imager and Imaging Melon Flowers

The X-ray computed tomography imager was made of an X-ray tube, an X-ray detector, an object rotation stage, and a computer. The X-ray tube irradiates the object under different angles, producing two-dimensional (2D) cross-sectional images [31]. The acquired images are then used to reconstruct the object in a 3D format (Figure 2a). The bioinformatic pipeline used for the image acquisition is shown in Figure 2b. We used micro-CT to obtain the scans of non-fixed male and female melon flowers. Flowers with peduncles were individually inserted in a water-filled microtube to avoid dehydration. The flower and the microtube were then placed into a glass probe and mounted in the microtomography unit (Figure 2a). With the standard setting, the first 3D rendering of individual CT slices resulted in a blurry output due to the vibration of the petals inside the apparatus, caused by the inner fan airflow. Nevertheless, adjusting the sample inside a polystyrene block was sufficient to avoid flower vibrations, generating a clear image. With this setting, the micro-CT imager allowed for the mounting and acquisition of the scans in about 30 min. We performed the scans at two resolutions. Scanning of whole male (Figure 3a,c) and female flowers (Figure 3b,d) was acquired in three adjacent CT slices at a resolution of 7.5 µm/px. Then, the scans were automatically merged with the NRecon Bruker software. The initial dataset of a single female flower contained more than 6000 files, representing about 124 Gigabytes (GB). However, after reconstruction and cropping, the dataset size was reduced to about 41 GB. The micro-CT 3D reconstruction of whole male and female flowers showed the entire external flower structure with the two common parts (corolla and calyx) and the hairy distinct one aborted ovary versus ovary. Cross-sectional views provided inner details on the sexual reproduction components and nectaries. Scans focused on the nectaries were acquired as a single CT slice, at a resolution of 5 µm/px (Figure 4c–j). The initial dataset of the focused scan contained 1029 files, representing about 20.7 GB. Reconstruction and cropping reduced the dataset size to about 4 GB for the male flowers and 7 GB for the female flowers. The image datasets, coming either from the micro-CT software or the Bruker 3D suite, were saved as images in a 16-bits tiff file format, and as such can be accessed using almost any image-processing software.

### 3.2. Phenotyping Nectar-Related Traits Using Micro-CT 

The flower shape, nectar volume, and nectary size are key traits controlling plant–pollinators interactions. However, the quantitative data on such traits are difficult to measure and 2D imaging techniques, such as a stereomicroscope, can be limiting and faulty (Figure 4a,b). To study the nectar-related traits of melon flower types, we used micro-CT (Figure 4c–j), which allowed for the imaging and measurements to be in 3D. In the 3D reconstructed images of melon flowers, we can identify four sets of floral organs arranged in concentric circles, namely the calyx, corolla, stamens, and pistils. In the staminate flower, the anthers almost completely fill the corolla tube (Figure 4c,e), whereas at the base of the corolla, in the center of the flower, the non-development of the pistil allows the development of nectary in the form of a dome (Figure 4g,i). In the female flower, the three-lobed stigma fills most of the corolla tube (Figure 4d,f). The style, carrying the stigma, is positioned at the center of the flower and is completely surrounded by the nectary (Figure 4h,j).

The flower width determines the physical access to nectar and as such represents an important nectar-related trait. We used micro-CT to measure the flower width in male and female flowers (Figure 4k; Appendix A). Female flowers were significantly wider (6.12 mm) as compared to the males (4.3 mm), corroborating the pattern described in the literature for other melon cultivars [5]. 

The nectary central cross-section area could be used as a proxy to estimate the nectary size. In a longitudinal section of the male flower, the nectary appears as a single organ at the center of the flower, while in the female flower, it appears as if two separated organs at each side of the pistil’s style. To calculate the nectary central cross-section area in a female flower, we have included both surfaces. Using micro-CT, we obtained a mean of 2.55 mm^2^ in male and 2.83 mm^2^ in female flowers, showing no significant difference between the studied flower types (Figure 4l, Appendix A). However, the nectary area is dependent on the angle of the longitudinal section of the micro-CT image. To accurately determine the nectar gland size, we used micro-CT to visualize and extract the total surface and nectary volume. In contrast to the nectary cross-sectional surface, the total outer surface in the female flowers was 2.5-fold higher as compared to that of the male flowers (Figure 4m, Appendix A). Similarly, the nectary volume in female flowers was 2.7-fold higher as compared to that of the male flowers (Figure 4n; Appendix A).

The floral morphological features have been studied in relation to the nectar production in various species and positive correlations were found between the amount of secreted nectar and the nectar gland size [9]. Similarly, in unisexual melon flowers, we observed positive correlations between the nectar volume secreted and the nectary size (Figure 5a–i, Appendix A), particularly with the nectar gland volume (Figure 5a,b; ♂: R^2^ = 0.71, r = 0.84, *p* < 0.01; ♀: R^2^ = 0.45, r = 0.67, *p* < 0.05) and the nectary area (Figure 5g,h; ♂: R^2^ = 0.67, r = 0.82, *p* < 0.01; ♀: R^2^ = 0.56, r= 0.74, *p* < 0.05), while the correlation between the nectar volume and nectary surface was only significant in the male flowers (Figure 5 d,e; ♂: R^2^ = 0.55, r = 0.74, *p* < 0.01; ♀: R^2^ = 0.40, r = 0.63, *p* > 0.05). In addition, in a pool of male and female flowers, the nectar volume was strongly correlated with the gland volume (Figure 5c; ♂♀: R^2^ = 0.73, r = 0.86, *p* < 0.0001) and nectar gland surface (Figure 5f; ♂♀: R^2^ = 0.79, r = 0.89, *p* < 0.0001). On the other hand, in unisexual flowers, no correlation was observed between the nectar volume and flower width (Figure 5j,k; ♂: R^2^ = 0.15, r = 0.39, *p* > 0.05; ♀: R^2^ = 0.18, r = 0.43, *p* > 0.05). Our findings are in line with the previous reports on nectary features. For example, a significant correlation (r = 0.98) between the nectar gland volume and the nectar volume per flower was reported for eight mint species [55]. Moreover, positive correlations between the nectar volume and floral features, e.g., the receptacle diameter, were reported in *Medicago sativa* [56]. Similarly, a strong positive correlation between the nectar quantity and volume of the receptacle was found in eight Lamiaceae species [55]. However, this was not the case for melon unisexual flowers (Figure 3, Appendix A).

Based on the previous studies as well as on our observations, nectar production is dependent on the physiological state of the plant at the moment of the measurement. We found a good correlation between the nectar volume and the different nectary attributes, indicating that the volume, the surface, and the cross-section area can be used as proxy to predict the nectar volume production.

### 3.3. Assessment of Nectar Accessibility to Honey Bees

As the nectar volume is positively correlated with the number of honey bee visits [57], we studied the total amount of secreted nectar. Owing to the morphology of a female flower which comprises an inner cavity between the nectary gland and the stigma (Figure 6a), a complete nectar removal from the intact flowers usually resulted in damaging the surrounding tissue. To ensure a complete removal of the nectar, we recorded the volume in both the intact (M1, F1) and dissected (M2, F2) flowers. In female flowers, a significant extra amount of nectar (6.9 µL) was collected from the inner cavity upon dissection (Figure 6a, Student’s *t*-test, *p* < 0.05; Appendix A). Similarly to the organ size, the nectar volume in female flowers (12.10 µL) was significantly higher than that of the male flowers (2.18 µL).

The recorded nectar volume (µL) was then used to project the nectar volume level inside each flower type, allowing a precise 3D overview of it (Figure 6c–f). To manipulate and process each 3D dataset, we used the Python language libraries. We defined a grey level threshold to select only the air-filled regions (the darker pixels) inside the flower. Using the linear image resolution of 5 µm/px, we calculated the volume represented by each voxel (3D pixel) according to the following formula: 1 voxel (Vox) = 1.25 × 10^−7^ µL (5 × 5 × 5 = 125 µm^3^)

Next, starting at the flower base and scanning upward, each voxel of the selected air-filled area was kept until reaching the recorded volume of the sampled nectar. This volume of interest is extracted and saved as a new 3D dataset, which can be displayed as an overlay on the initial greyscale images.

To reconstruct a honey bee visitation event, we first acquired 3D images of the honey bee morphology (Figure 7a) and interior organs (Figure 7b). In the morphology scan, the topological and geometric properties of the abdomen, thorax, and the head are easily identified. In the anatomy scan, the heart, the honey sac (crop), the flight muscles, the brain, and the proboscis can be easily distinguished (Figure 7b). To ease the manipulation of the 3D dataset, we made a subsample of the 3D rendering by selecting only the honey bee’s head. To assess the flower accessibility, we performed a modeling of the honey bee visitation by superimposing three 3D datasets: the honey bee, melon flower, and the reconstructed nectar volume. The honey bee’s head was placed deep into the flower (Figure 7c,d), depending on the available space between the petals and the surrounding floral organs. This resulted in a plausible position in which the honey bee’s proboscis easily reaches the bottom of the flower. 

## 4. Discussion

In most fruit crops, breeders have shaped the flowers to improve the fruit morphology, but how this selection impacted the nectar production and accessibility for bees is unclear. Today, despite the need to enhance the crop-ecological functions and prevent further pollinator decline, breeding flowers to enhance the pollination services is still far from feasible due to the difficulty of phenotyping nectar-related traits. In rare published works, researchers compared the floral characteristics in relation to the nectar characteristics and the frequency of visits of a given species of pollinators. Rudimental equipment such as digital calipers and rulers were used to measure the corolla diameters, flower height, and nectar chamber width [5,13,18,21,22]. 

In this study, we evaluated the use of micro-CT as a non-destructive tool for improving the imaging resolution and phenotyping quality of the nectar-related traits. Previous micro-CT studies used a fixation protocol, which includes dehydration and staining, prior to the micro-CT scanning of plant tissues [31]. Most microtomographic studies use an additional infiltration step with a contrasting agent such as phosphotungstic acid (PTA) for several days. This chemical compound, associated with ethanol can improve the visualization of specific internal flower tissues and cells which are not in direct contact with the air, such as ovaries [58] or vascular bundles of flowers [59]. To the best of our knowledge, only two studies were published with similar scanning conditions without any contrasting agent concerning the leaves [31,50].

In our investigation, we show that a non-fixed strategy is rapid and sufficient to generate high-quality 3D scans. This simplified protocol has the advantage of guaranteeing the native integrity of the shapes and volumes of the floral organs in contact with the air, rendered by the microtomographic analysis. The main challenge we encountered during the analysis of flower datasets is the residual nectar left inside the flowers prior to the scanning. If a significant amount remains inside the flower, it hinders the identification of the boundaries between the individual micro-CT slices, altering the organ morphology assessment during the 3D image processing. 

The investigation of the two flower types by micro-CT allowed for the visualization of the surface and internal floral features (Figure 2 and Figure 3). Surface organs such as trichomes on the calyx can be easily revealed and quantified [60,61,62]. Flower trichomes have been associated with the attraction of pollinators, the repulsion of non-winged insect pests, or the regulation of the temperature inside the flower [63]. As floral shape variation often reflects the differences in plant pollination systems, particularly for plant species with specialized plant–pollinator interactions [64], studying floral geometric measures using micro-CT can be instrumental not only for flower development, but also for plant–pollinator co-evolution.

In terms of 2D imaging, classical stereomicroscopy analysis requires a manual dissection of the flower, and as such the phenotyping result is highly affected by the sectioning plane (Figure 3). In contrast, flowers observed by micro-CT remain intact and can be analyzed in silico under different angles. In our case study, the structural comparison of the sectional width of melon unisexual flowers showed that male flowers are one third smaller than female flowers. This difference in size correlates with the volumes occupied by the sexual organs and the nectar gland inside the flower (Figure 3). 

In terms of 3D imaging, the microtomographic results clearly showed significant differences in both the nectary surface area and the nectary volume between male and female flowers, showing bigger female nectaries. In comparison with the 2D analysis, the 3D definitely allows a more precise quantification of this type of flower organ and highlights the added value of the microtomography technique and image processing for these types of 3D measurements.

In addition to providing more structural details, the 3D analysis brings other important advantages such as capturing the additional traits, e.g., the corolla curvature that cannot be observed using the conventional 2D methods [23]. The corolla curvature has been demonstrated to act as a mechanical nectar guide, which facilitates the direct flower handling for plant–pollinator interactions [65]. Similarly, a phenotypic analysis of developmental mutants with irregular organ morphologies or wild accessions with slightly altered shapes could be significantly facilitated using 3D micro-CT. 

Image processing is a powerful tool to obtain specific 3D information of different organs and structures. To facilitate data analysis for high throughput phenomics, landmarks can be detected automatically without user intervention by the newly developed algorithms [29]. However, as previously reported, the challenge that prevents us from automating organ measurements using machine learning is the lack of an X-ray absorption difference between the neighboring structures and thus the lack of color difference; for example, between the adjacent floral tissues or between a flower and an embedding medium used to preserve its 3D form [33,66]. In both cases, the challenge can be overcome by applying advanced scanning technologies such as a phase contrast synchrotron X-ray tomography [67,68] which can enhance the contrast at the interface between the two media of a similar density, but this requires a synchrotron facility.

Beside the plant, we showed that micro-CT can be used for scanning honey bees, which together with scanned flowers can serve to visualize the nectar volume inside the nectar chamber, the bee’s accessibility, and to quantify and model plant–pollinator interactions (Figure 4). Even though the 3D image of the flower–bee interaction was obtained from the superposition of two images and not a single scan, the extracted information permits to investigate the compatibility of a pollinator with a given flower. Once a 3D image of a pollinator and a flower have been generated, we can generate in silico a library of flower shapes that mimic the diversity of flower morphologies in a given species. Plants displaying flower shapes superposing in silico flower models could be then tested for compatibility or incompatibility with pollinator visitation. The investigated parameters could include access to nectar and pollen harvesting and the deposition on the stigma (Appendix A).

## 5. Conclusions

The use of micro-CT as a tool for floral organ phenotyping represents an excellent compromise between the classical imaging and the synchrotron facility. On the one hand, as a non-destructive, high-precision imaging tool, it allows a rapid acquisition of 2D and 3D images at the resolution of the cellular. We scanned the intact melon flowers without a tissue fixation and thus shortened the protocol for the acquisition of high-quality 3D images. The resulting scans were used for obtaining geometric measurements of organ sizes and nectar volumes as well as for the respective correlation analyses. The main advantage of this method is the reduced acquisition time for obtaining high-resolution 3D data which provides a higher precision compared to the 2D images in phenotyping floral morphology. We believe that the application of the micro-CT technique has a great potential to improve the interdisciplinary studies of plant–pollinator interactions. Its relative low cost and time efficiency will facilitate a fast collection of high-quality phenotyping data. The established protocol will permit to readily carry out a comparative analysis of a phenotype–pollinator interaction and identify nectar-pollination-related traits while guaranteeing the integrity of native shapes and volumes of the floral organs.

## Figures and Tables

**Figure 1 cells-11-03452-f001:**
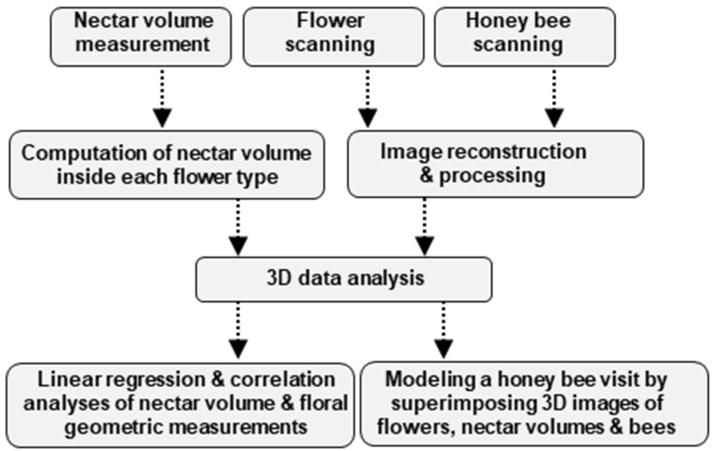
Mind map of the methodology used in this study. Three independent samplings were carried out, the sampling and measurement of nectar and scanning of the flower and the honey bee. Three-dimensional volumetric data was obtained for the three samples which were used for geometric measurements of organs of interest. Finally, correlation analyses were performed between nectar volume and different nectary attributes. We modeled a honey bee visit by superimposing three data sets: the nectar volume, flower, and the honey bee.

**Figure 2 cells-11-03452-f002:**
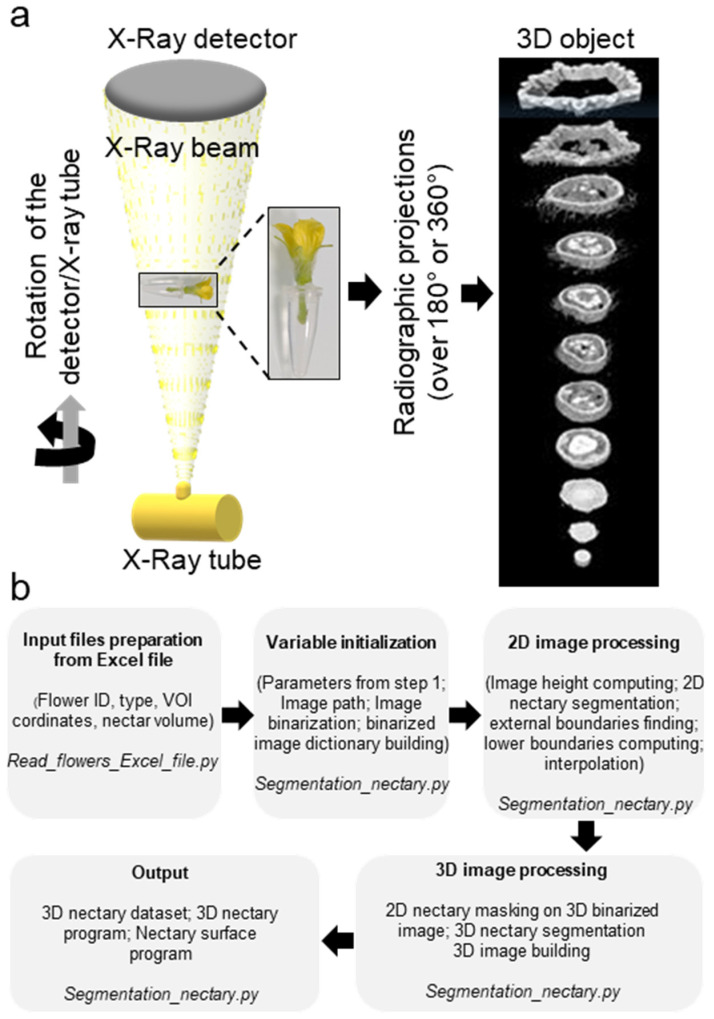
Microcomputed tomography of melon flowers at anthesis stage. (**a**) Schematic diagram of an in vivo microcomputed tomography device. The X-Ray tube generates an X-rays conical beam. The sample, the flower, was placed inside a polystyrene chip and inserted into a water-filled microtube. The beam crossing the sample was collected by the X-ray detector leading to radiographic projections. Then, the projections were used to compute the three-dimensional structure of the sample. (**b**) Schematic representation of the bioinformatic pipeline used for acquisition of the 3D images.

**Figure 3 cells-11-03452-f003:**
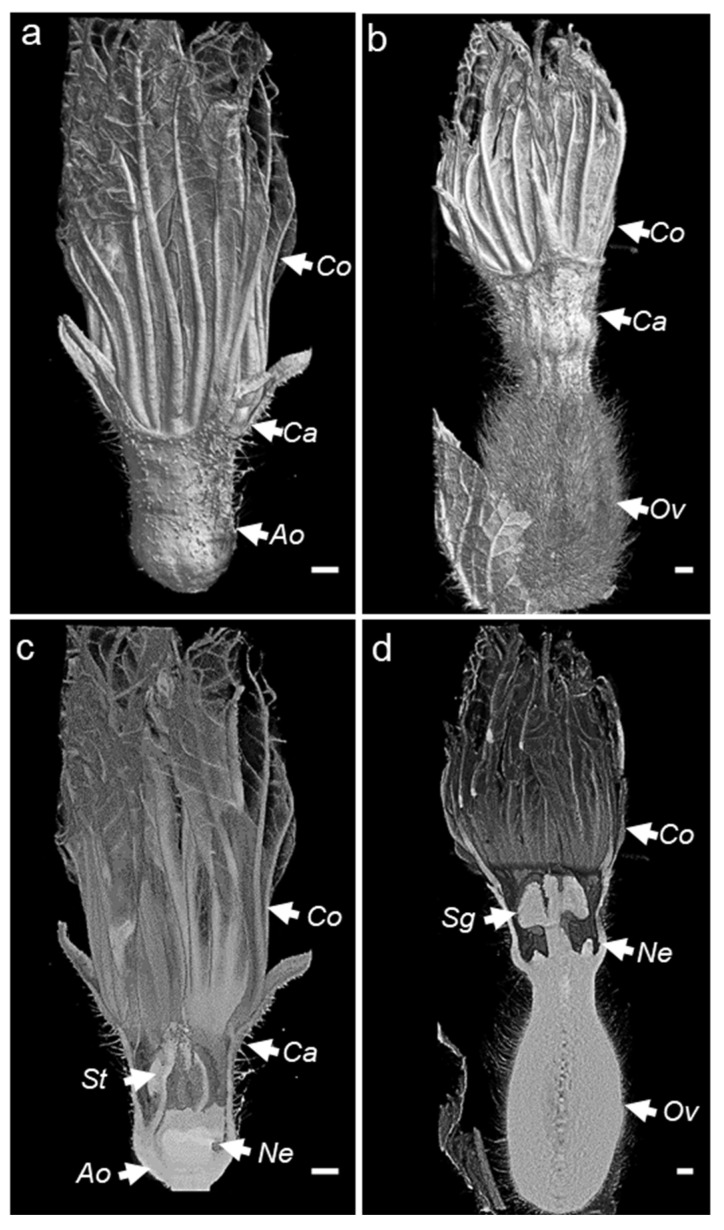
(**a**–**d**) External and cross-sectional 3D overview of male (**a**,**c**) and female (**b**,**d**) melon flowers generated using Bruker CTvox software. Ov, ovary; St, stamen; Sg, stigma; Ne, nectary; Co, corolla; Ca, calix; Ao, aborted ovary; VOI, volume of interest. The scale bar represents 1 mm.

**Figure 4 cells-11-03452-f004:**
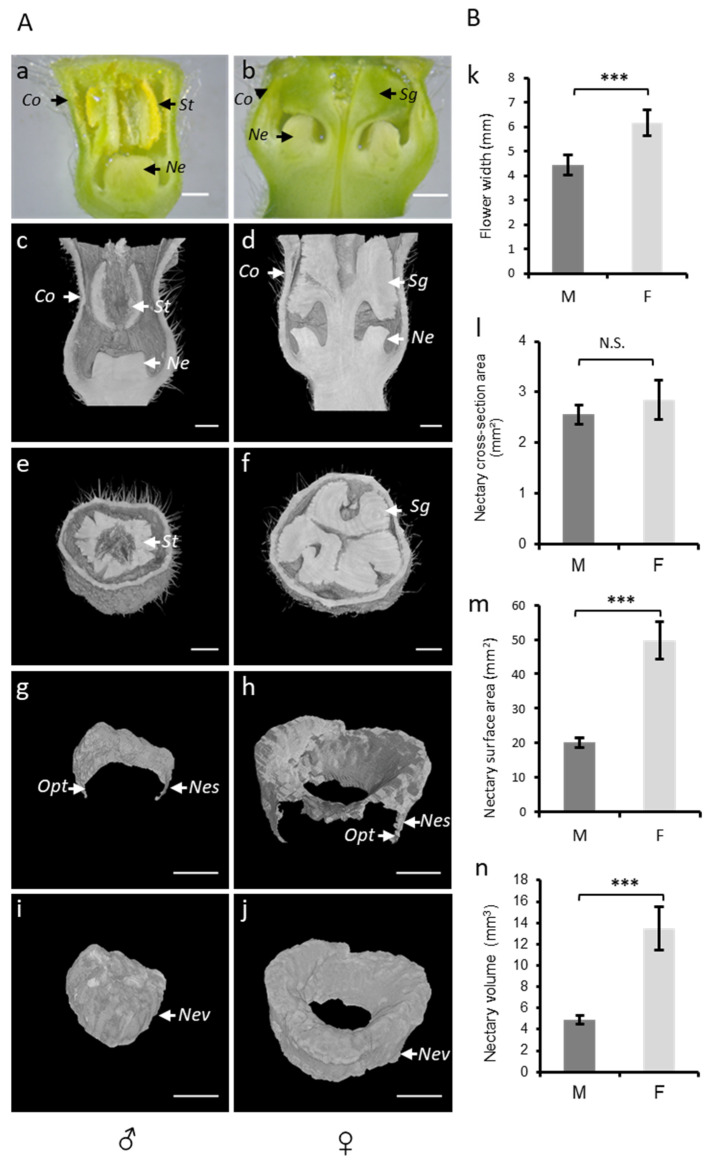
(**A**) Analysis of melon flowers at anthesis stage using a stereomicroscope and microtomography. Transversal view of ♂ (**a**) and ♀ (**b**) flowers under a stereomicroscope. Microtomographic transversal and sagittal views of ♂ (**c**,**e**) and ♀ (**d**,**f**) flowers. Three-dimensional cross sectional views of total 2D nectary surface area (**g**,**h**) and 3D views of total nectary volumes (**i**,**j**) extracted from 3D models of flowers types (♂, ♀). St, stamen; Sg, stigma; Co, corolla; Opt, one pixel thickness; Ne, nectary; Nes, nectary surface; and Nev, nectary volume. The scale bar represents 1 mm. (**B**) comparison of 2D and 3D parameters between flower types (♂, ♀): flower width (**k**), cross-sectional nectary area (**l**), total 2D nectary surface area (**m**), and nectary volume (**n**) of ♂ and ♀ flowers calculated using microtomography. Male = M; female = F.; N.S. = not significant; *** = *p* < 0.001.

**Figure 5 cells-11-03452-f005:**
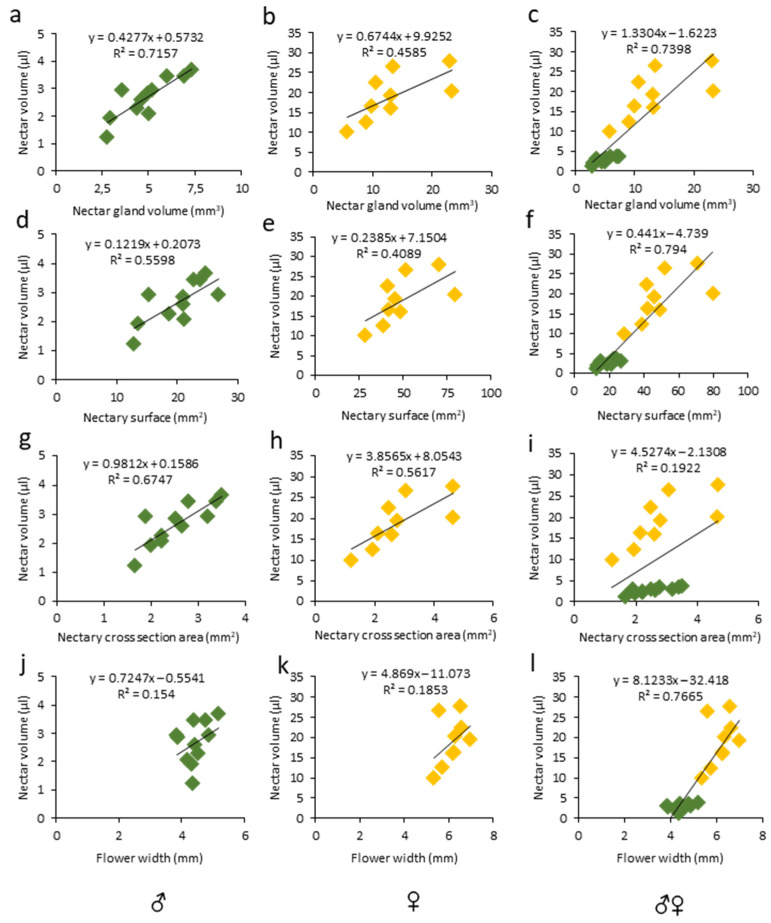
Linear regression and correlation analysis between nectar volume and (**a**–**c**) nectar gland volume, (**d**–**f**) 2D nectary surface, (**g**–**i**) nectary cross-section area, and (**j**–**l**) flower width in the respective male, female, and pooled melon flowers. R^2^ coefficients with equations are shown above each slope. Number of samples (n) was 11 and 9 for male (green) and female (yellow) flowers, respectively.

**Figure 6 cells-11-03452-f006:**
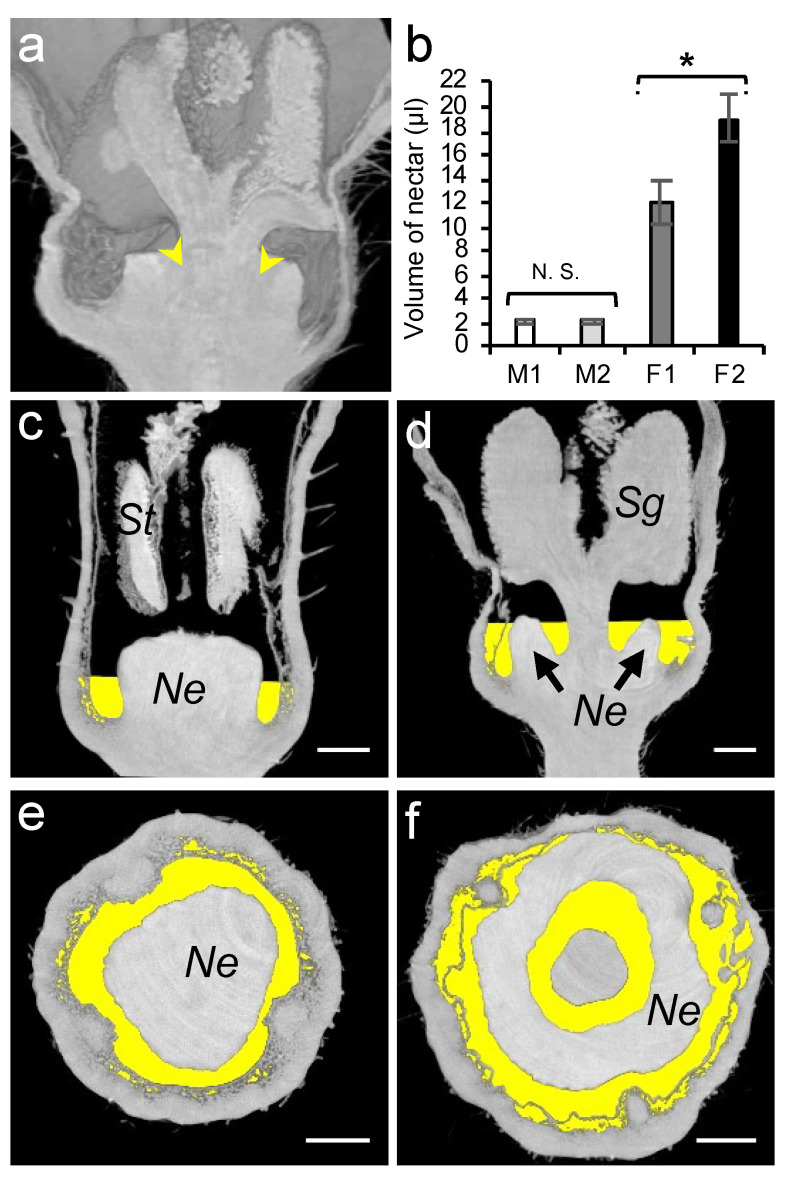
Analysis of nectar volumes of male and female melon flowers at anthesis stage. (**a**) Nectar left in female flower after glass capillary pipetting is indicated by yellow arrows. (**b**) Nectar volume in male (M) and female (F) flowers measured using glass capillary pipettes with (M2 and F2) and without flower dissection (M1 and F1). St, stamen; Sg, stigma; Ne, nectary; and N.S., not significant; * = *p* < 0.05. Microtomographic transversal and semi-sagittal views of male (**c**,**e**) and female (**d**,**f**) flowers showing nectar volumes in yellow. The scale bar represents 1 mm.

**Figure 7 cells-11-03452-f007:**
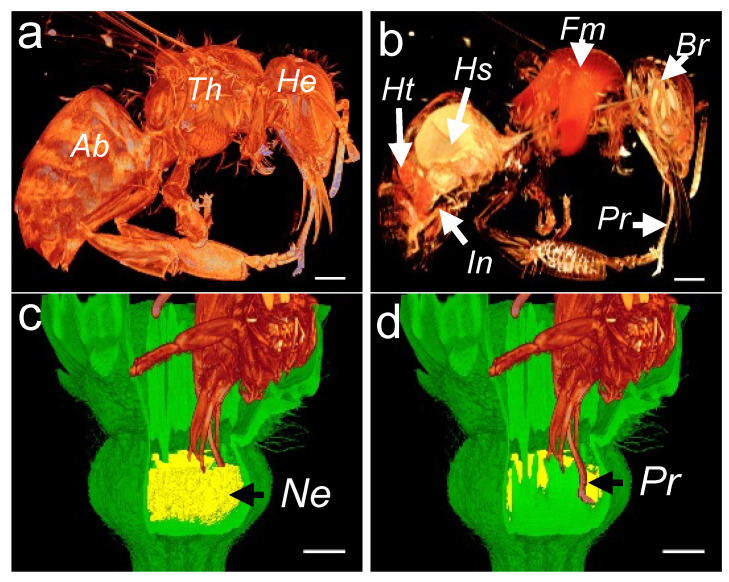
Reconstruction of 3D images showing the head of a honeybee sucking nectar. External (**a**) and cross-sectional (**b**) 3D images of a honeybee. Three-dimensional image showing nectar volume (**c**) and honeybee proboscis with the sliding glossa (tongue) pumping nectar deep into the flower corolla (**d**). Ab, abdomen; Th, thorax; He, head; Ht, heart; Hs, honey stomach; Fm, flight muscles; Br, brain; In, intestine; Pr, proboscis and glossa; and Ne, nectary. The scale bar represents 1 mm.

## Data Availability

The bioinformatics pipeline is deposited in the GitHub at https://github.com/El-Castor/X-ray-Micro-computed-Tomography.

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
