# Peer review of "Precision Phenotyping of Nectar-Related Traits Using X-ray Micro Computed Tomography"

_cells, 2022, doi:10.3390/cells11213452_

Round 1

Reviewer 1 Report

Line 32-33: “Flower morphology, together with color and scent, is one of the key traits that 32 contribute to pollinator attraction, pollen dispersal and plant reproductive success” please provide citations.

Line 37: The genus name should italic.

Line 49: Would be better to provide scientific  name.

Line 50: The Genus name should be italic.

Line 56: “Cucurbita pepo” Species name should be italic.

Line 84: The authority of the species should not be italic.

Line 98: same as above

Line 108-116: subheadings; “2.2. Micro-CT scanning conditions” which methodology the authors followed? Please provide citations.

Same for the subhedings 2.4. Honey bee preparation

Line 2.8: Would be better to provide the detail of package used during analysis in R as well.

Line 258-270: Please recheck and confirm the p-values provided and compare with the figures

Line 353-357: “Investigation of the two flower types by micro-CT allowed the visualization of sur- 353 face and internal floral features (Fig. 1). Surface organs such as trichomes on the calyx can 354 be easily revealed and quantified [47-49]. Flower trichomes have been associated with at- 355 traction of pollinators, repulsion of non-winged insect pests or the regulation of 356 c d a b Ne Pr Pr Th He Ab Hs Fm Br Ht In Cells 2022, 11, x FOR PEER REVIEW 13 of 16 temperature inside the flower [50]. Investigation of their shapes, their density and their 357 types can be instrumental in the field of agroecology” Please modify this paragraph.

Line 373-377: “Image processing is a powerful tool to isolate and extract specific 3D organ infor- 373 mation from scans. To facilitate data analysis for high throughput phenomics, landmarks 374 can be detected automatically without user intervention by the newly developed algo- 375 rithms [26]. However, as previously reported, the challenge that prevents us from auto- 376 mating the organ measurements using machine learning is the lack of color difference 377 between the tissues/organs [52].” This paragraph is confusing please clarify the statement and add more data in this paragraph.

Please check the Rvalues in all the figures. For instance; "R² = 0,7665" or 0.7665? 

Please add recommendations as well.

The discussion section is weak and needs improvement please discuss and compare your results with previously published literature.

Please check for minor grammatical mistakes throughout the ms.

Arrange the references as per the guidelines of the journal. 

Reviewer 2 Report

This is a lovely little paper, showing that X-ray Micro CT can be used to look at flower structure and nectary space in a short 30 minute scan without the requirement of fixing. This therefore makes this method a relatively simple process to look at easy to determine factors (e.g. sizes). As mentioned in the paper it has limitations at the tissue level where you are unable to easily separate different tissues based on a histogram as they all have a similar density and therefore would require some sort of fixation or computer analysis to determine exact tissues if doing a large scale phenotyping that required that detail. All in all this is a nice novel technique to look into nectar-related traits, and may provide a way to do in situ modelling of insect and flower behaviour based on shapes.

The paper is well written with the sort of methods and analysis I would expect from an X-ray MicroCT analysis, and therefore I only have a few minor comments;

Figure 1: Line 209 – From my understanding the microcentrifuge tube and sample were further placed into a polystyrene block to prevent air movement. Was this placed around the sample completely or just as a barrier for the air flow? It would be good to include this within the image. As without this you suffer from the distortions of movement and airflow during the scan itself.

Figure 2: Line 234 – Ne is not described.

References: I can only mention the X-ray CT side as I do not have expertise in the pollination. But they are missing a few references of work that is similar (but different species) that I would expect to be mentioned in the introduction and/or discussion. Articles where they have come across similar problems, and/or studied similar things E.g.

van der Niet T, Zollikofer CP, León MS, Johnson SD, Linder HP. 2010. Three-dimensional geometric morphometrics for studying floral shape variation. Trends in Plant Science 15, 423–426.

Ijiri T, Yoshizawa S, Yokota H, Igarashi T. 2014. Flower modeling via X-ray computed tomography. ACM Transcations on Graphics 33, 48.

Tracy SR, Gómez JF, Sturrock CJ, Wilson ZA, Ferguson AC. 2017. Non-destructive determination of floral staging in cereals using X-ray micro computed tomography (µCT). Plant Methods 13, 9.

Staedler YM, Kreisberger T, Manafzadeh S, Chartier M, Handschuh S, Pamperl S, Sontag S, Paun O, Schönenberger J. 2018. Novel computed tomography-based tools reliably quantify plant reproductive investment. Journal of Experimental Botany 69, 525–535.

Finally;

Supplemental 3D videos and information. While I understand the limitations of storage, it would be nice to have the information/slices generated here available for public use, similar to the storing of RNA-seq data on a public platform. At the very least a 3D video of a male/female flower and the bee would be a good starting point for people to understand what they can really generate from the X-ray CT analysis. Having the data available would allow other traits to be analysed and compared, for example positioning of the anther during an insect/bee visit.

Reviewer 3 Report

The article is devoted to the use of microcomputed tomography to build a 3D model of flower availability for honey bees. The material is of interest and novelty, but there are the following remarks.

1) In the "Introduction" section. Write, what specific problems were not solved in the above studies using the CT method? (when describing literary sources 29-36) So what is the novelty of your research compared to those published earlier?

2) At the end of the "Introduction" section, you must write the purpose of the study and objectives

3) At the beginning of the Materials and Methods section. The methodology should be improved by adding mindmaps/graphs and further clarifications.

4) In Section 2 "Materials and Methods" it would be necessary to present a block diagram of a complex of technical means (from sensors/cameras to a computer) and the structure of the software and the functions that they perform.

5) Section 2 should add the calculation and statistical formulas that are used

6) It is advisable to divide Figure 1 into 2 figures - a separate figure - a diagram, a separate figure with the results of image processing. Make vertical inscriptions in one direction (“Radiographic……” place from bottom to top)

7) Please provide a separate figure with a fragment of the input Excel file.

8) In section 3 "Results". Please write down the practical significance of the obtained correlation dependences shown in Fig. 3. How can they be used in the future?

9) The conclusions should be structured and tied to the tasks of the work.

10) Also in the conclusions it is necessary to show the superiority of the proposed methods in comparison with the existing ones (for example, in percentage, which indicator has improved how much). Accordingly, in the "Results" section, this should be justified.

Round 2

Reviewer 1 Report

The authors respond well to all the suggested queries. Now the quality of the manuscript is much improved, easy to read and understandable. No, need further changes therefore recommended for publication in its present form.

Reviewer 3 Report

I am generally satisfied with the answers of the authors and the corrections made by the authors to the article in response to my comments